# HUMAN-IN-THE-LOOP DETECTION OF AI-GENERATED TEXT VIA GRAMMATICAL PATTERNS

## ABSTRACT

The increasing proliferation of large language models (LLMs) has raised significant concerns about the detection of AI-written text. Ideally, the detection method should be accurate (in particular, it should not falsely accuse humans of using AI-generated text), and interpretable (it should provide a decision as to why the text was detected as either human or AI-generated). Existing methods tend to fall short of one or both of these requirements, and recent work has even shown that detection is impossible in the full generality. In this work, we focus on the problem of detecting AI-generated text in a domain where a training dataset of human-written samples is readily available. Our key insight is to learn interpretable grammatical patterns that are highly indicative of human or AI written text. The most useful of these patterns can then be given to humans as part of a human-in-the-loop approach. In our experimental evaluation, we show that the approach can effectively detect AI-written text in a variety of domains and generalize to different language models. Our results in a human trial show an improvement in the detection accuracy from $43\%$ to $86\%$, demonstrating the effectiveness of the human-in-the-loop approach. We also show that the method is robust to different ways of prompting LLM to generate human-like patterns. Overall, our study demonstrates that AI text can be accurately and interpretably detected using a human-in-the-loop approach.

## 1 INTRODUCTION

Large language models (LLMs) are demonstrating exceptional capability across diverse domains, including logical reasoning, fluent language usage, and comprehensive factual awareness (Brown et al., 2020; Chowdhery et al., 2022). These capabilities bring new risks such as a tendency to hallucinate new information (Bang et al., 2023), introduce biases (Liang et al., 2021a), violate privacy (Brown et al., 2022), and others. One of the most widely discussed consequences is the lack of ability to distinguish between human and AI written text. This is an important problem as these models can disseminate misinformation at a large scale which can threaten democracy and trust in institutions (Chee, 2023; Juršėnas et al., 2021; Azzimonti & Fernandes, 2023) and propagate biases (Ferrara, 2023; Liang et al., 2021b). This is not just a future concern, as we have already seen the use of AI to generate Amazon product reviews (Palmer, 2023) or write novels for magazines (Hern, 2023). Moreover, educational institutions are also expressing concerns that traditional approaches for measuring student performance are not so effective in the face of new technology.

This underlines the importance of having a reliable and accurate way of identifying text written by AI. However, existing research falls short of this goal, as the proposed methods either only work on smaller models (Mitchell et al., 2023), require integration of detector and text generation(Kirchenbauer et al., 2023), or generally have low accuracy. Importantly, all of these methods do not provide an explanation of why some text has been classified as AI, as the decision is based on probabilities computed using deep neural networks, which are highly non-interpretable. This can have important negative consequences, as low accuracy and non-interpretability mean that a large number of innocent people will be accused of submitting AI-written text while receiving no explanation for this decision.

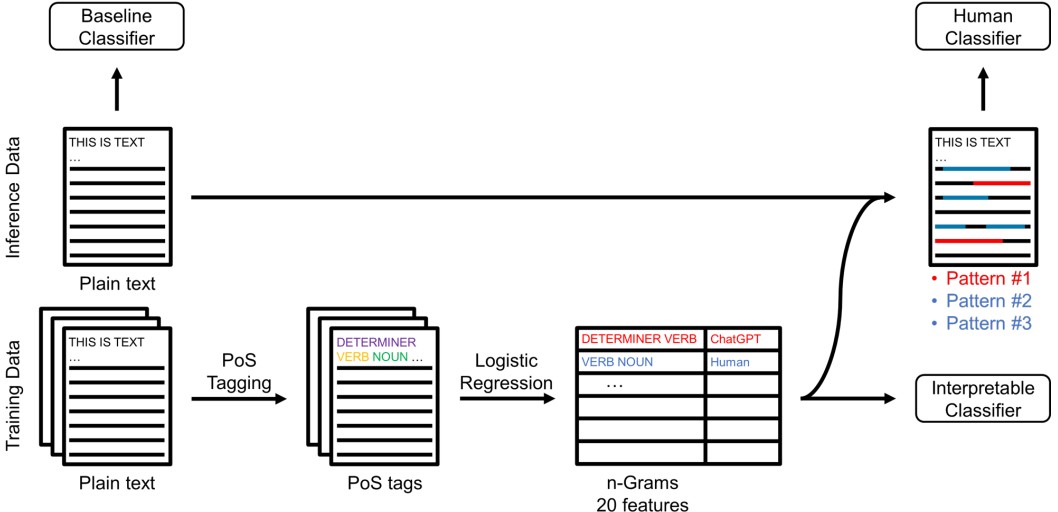

Figure 1: Overview of our approach. We perform PoS tagging on the training dataset and learn the most frequently occurring patterns in both human (blue) and LLM (red) written text. We then use these patterns to either train an interpretable classifier or give them directly to a human to assist in the detection process, thus creating a human-in-the-loop approach.

**This work**    In this work, we propose a novel method to detect AI-written text using machine learning with a human-in-the-loop approach, which allows us to achieve high accuracy and interpretability. As the problem of detection has been shown to be intractable in the most general setting (Sadasivan et al., 2023), we focus on a case where there is an available dataset of human written text from a certain domain (e.g. arXiv texts or news articles). An overview of our method is shown in Figure 1. We first supplement the dataset with texts produced from a large language model by prompting it to generate texts from the same domain (e.g. abstracts of scientific articles with given titles). While we could train a highly accurate deep neural network classifier (e.g. by fine-tuning the BERT model) for detection, it would still not solve the problem of interpretability. Instead, our key insight is to learn PoS (part of speech) patterns whose occurrence is highly predictive of either human or AI-written text. To achieve this, we extract the number of occurrences of each PoS pattern in the text and then train the logistic regression classifier to discriminate between human and AI written text using these PoS features. Finally, we select the 20 patterns whose features have the highest (indicative of human) or lowest (indicative of AI) weight.

In our experimental evaluation, we show that our approach is effective. When humans are assisted with our patterns, their accuracy increases from 40% (worse than random guessing) to around 86% (close to a non-interpretable ML classifier). We also experimented with changing the prompt to LLM so that it generates more human-like patterns and showed that our approach still performs well, demonstrating that it is quite robust. These results indicate that interpretable AI detection with a human-in-the-loop approach can lead to classification with high accuracy and interpretability.

**Main contributions**    Our main contributions are:

- We propose a novel method for accurate and interpretable detection of AI-generated text using a human-in-the-loop approach.

- In our experimental evaluation we demonstrate that the method generalizes across different state-of-the-art LLMs and text domains and is robust against several evasion strategies.

- We demonstrate the practical utility of our method through a human trial where results indicate that we can enable non-experts to identify machine-generated texts.

## 2 RELATED WORK

Several approaches have emerged to address societal concerns about the increasing difficulty of identifying LLM-generated texts. Current techniques differ in several key aspects. One pertains to output manipulation, where detection is simplified by modifying LLMs to embed characteristic signals in the produced text (Mitchell et al., 2023). Another important distinction involves accessing the model's probability distribution: white-box methods require this access, whereas black-box methods do not (Tang et al., 2023). Moreover, approaches also bifurcate into zero-shot techniques and those reliant on training sets for effective generalization to new contexts. Additionally, detection methods vary in terms of interpretability. These attributes reveal the main strengths and limitations of existing techniques.

**Learned classifiers** have been trained to distinguish human-authored and LLM-generated texts. In particular, BERT-based architectures like RoBERTa and DistilBERT (Devlin et al., 2019; Liu et al., 2019; Sanh et al., 2020) have been fine-tuned to accurately identify scientific abstracts produced by ChatGPT (Guo et al., 2023; Mitrović et al., 2023; Wang et al., 2023b; Yang et al., 2023; Yu et al., 2023; Theocharopoulos et al., 2023). Various training strategies such as contrastive or adversarial learning have also been successfully applied to increase performance (Bhattacharjee et al., 2023; bing Hu et al., 2023; Koike et al., 2023). Moreover, to foster human-AI interaction, post-hoc explanation methods such as SHAP and Polish-Ratio have been studied (Lundberg & Lee, 2017; Mitrović et al., 2023; Yang et al., 2023). Others have suggested reducing model complexity, for instance through gradient boosting tree classifiers reliant on linguistic features (Desaire et al., 2023).

**Zero-short methods** do not require training data, in contrast to learned classifiers. Recently, statistical tests have been developed to detect texts produced by a specific LLM; often using pivots based on token-wise conditional probabilities such as average token log probability, mean token rank, and predictive entropy (Gehrmann et al., 2019). Similarly, DetectGPT relies on a curvature-based criterion for zero-shot detection (Mitchell et al., 2023). The curvature is estimated through multiple perturbations of the original text, using a partial masking approach. Su et al. (2023) expounds upon similar ideas, utilizing log-rank information and specifically normalized log-rank perturbations.

**Watermarking** is the most prominent attempt of making detection less challenging (Kirchenbauer et al., 2023) and has recently garnered endorsement from major tech companies as well as the US government as a safeguard against AI-misuse (Bartz & Hu, 2023). This technique partitions output tokens into distinct green- and red-list categories, compelling the model to predominantly generate green-listed tokens. Applying statistical convergence results, one can ensure accurate identification of LLM-generated texts, accompanied by statistical guarantees on the false-positive rate. Moreover, recent works have improved the robustness, information content, and textual integrity of the watermark signal (Zhao et al., 2023; Wang et al., 2023a; Kuditipudi et al., 2023; Christ et al., 2023).

**Human ability** to recognize LLM-generated texts and methods to improve it has also been investigated. Several works have shown that without guidance, humans struggle to recognize LLM-generated texts (Gehrmann et al., 2019). However, by employing the visualization tool GLTR, peoples' accuracy when identifying GPT-2-generated texts increases from 54% to 72% (Gehrmann et al., 2019). Still, for current state-of-the-art LLMs, the performance is significantly lower (Uchendu et al., 2021). Moreover, mixed-initiative approaches aimed at enhancing experts' ability to recognize LLM-generated content have also shown great potential, with current work analyzing and visualizing syntactical, semantical, and pragmatical features (Weng et al., 2023). Also, collaboration between humans can increase their ability to collectively recognize LLM-generated texts, but without further guidance, the accuracy is still below 70% (Uchendu et al., 2023).

Moreover, as the quality of LLMs continues to improve and their text-generation capabilities approach that of humans, distinguishing between human and LLM-generated texts becomes increasingly challenging. In a recent study (Sadasivan et al., 2023), an upper limit on detection performance was established, which is based on the total variation distance between probability distributions of human authors ($\mathbb{P}_{\mathcal{H}}$) and the investigated LLM ($\mathbb{P}_{\mathcal{M}}$). Specifically, the area under the receiver operating characteristic curve (AUROC) of any classifier is upper bounded by:

$$\text{AUROC} \leq \frac{1}{2} + \text{TV}(\mathbb{P}_{\mathcal{M}}, \mathbb{P}_{\mathcal{H}}) - \frac{1}{2}\text{TV}(\mathbb{P}_{\mathcal{M}}, \mathbb{P}_{\mathcal{H}})^2 \tag{1}$$

Nevertheless, even though detecting machine-generated texts is a challenging problem, it carries immense societal importance.

## 3 BACKGROUND

In this section, we introduce the necessary background for our work.

**Problem formalization**  AI Detection is a binary classification problem concerned with discerning human-written and LLM-generated texts. Let $\mathbb{P}_{\mathcal{H}}$ and $\mathbb{P}_{\mathcal{M}}$ be the distribution of texts authored by humans and the investigated LLM, respectively. Text-label pairs are then sampled $(t, y) \sim \mathbb{P}$ with $y \sim \text{Unif}(\{0, 1\})$, $\mathbb{P}[\cdot \mid y = 0] = \mathbb{P}_{\mathcal{H}}$ and $\mathbb{P}[\cdot \mid y = 1] = \mathbb{P}_{\mathcal{M}}$. The problem of AI detection is then to construct a classifier $f_{\theta} : \mathcal{T} \rightarrow \{0, 1\}$ that accurately predicts author $y$ (human or AI) given text $t$ where $\mathcal{T}$ is the set of all texts.

**Grammatical patterns**  Grammar specifies which word sequences are contained in a language, and provides syntactic rules for how words can be combined into sentences. In most languages, these are formulated based on parts-of-speech (PoS) or word classes (Kroeger, 2005). Modern English employs nine fundamental word classes, as depicted in Table 1. Furthermore, the problem of assigning the appropriate PoS tag to each word is challenging due to polysemy and context dependency and has been extensively studied in computational linguistics. Modern approaches often employ machine learning and rely on a hidden Markov assumption (Toutanova et al., 2003; Toutanvoa & Manning, 2000; Zewdu & Yitagesu, 2022). Moreover, the resulting sequence of PoS tags contains all grammatical information from the original text. For illustration, we provide an example of the mapping between plain text and the sequence of PoS tags given in Table 1:

This is a sentence $\implies$ DETERMINER VERB DETERMINER NOUN

**Feature Selection**  In many machine learning applications feature selection, aimed at identifying the most informative attributes while discarding irrelevant or redundant ones, is a crucial step. Reducing the number of active features can boost model performance and interpretability. In particular, a large feature set limits comprehensive model understanding (Miller, 1956). Moreover, achieving optimal feature selection is known to be NP-hard, even in linear scenarios (Welch, 1982). As a result, heuristic approaches are often employed, generally extracting or combining attributes based on various heuristically motivated criteria (Jolliffe, 2002; Hyvärinen & Oja, 2000; Peng et al., 2003).

In this work, we perform feature selection using Lasso (Tibshirani, 1996). Specifically, sparsity is induced by applying $l_1$-regularization to the optimization problem

$$\arg\min_{\theta \in \mathbb{R}^n} \mathcal{L}(\theta) + \alpha \|\theta\|_1 \tag{2}$$

where $\mathcal{L}$ is the loss function and $\alpha$ is the regularization parameter. Sparsity generally increases as $\alpha > 0$ grows, and it can be shown that for any feature there exists an upper bound on the value of $\alpha$ such that it is still contained in the active feature set (Henche, 2013; Tibshirani, 1996). Therefore, by adjusting $\alpha$ one can approximate the optimal active feature set.

## 4 OUR METHOD

In this section, we introduce key parts of our method.

**Formalization**  Currently, humans struggle to recognize LLM-produced texts, often because they don't know which information is relevant. This issue is addressed by the following general framework, allowing for synergistic human-AI cooperation. Using a training set of text-label pairs, $\mathcal{D} = \{(t_i, y_i)\}_{i=1}^n$, we learn a function, $h_{\phi}$, that highlights certain text passages accompanied by an explanation of their relevance regarding the origin of the text. Formally, we construct

$$h_{\phi} : \mathcal{T} \longrightarrow (\mathcal{T} \times \mathcal{T})^*, t \mapsto \{(p_i, e_i)\}_i \tag{3}$$

where $\{(p_i, e_i)\}_i$ are pairs of highlighted text passages and corresponding justifications. These are provided to a human decision-maker who contextualizes the information and weighs the arguments against each other, before making the final decision.

| | Word Class | Definition |
|---|---|---|
| **Word Classes** | NOUN | A reference to a person, place or thing |
| | VERB | A reference to an action |
| | ADJECTIVE | A description of a noun's properties |
| | ADVERB | A description of a verb's properties |
| | PRONOUN | A substitute for a noun and any words which depend on it |
| | INTERJECTION | An expression that occurs as an utterance |
| | PREPOSITION | An description of a relationship in space or time |
| | CONJUNCTION | A link between different clauses of a sentence |
| | DETERMINER | A reference to a noun and any words which depend on it |
| **Penn Treebank Extensions** | DIGIT | A number or digit |
| | MODAL | An auxiliary verb expressing necessity, possibility, or permission |
| | EXISTENTIAL THERE | The word "there" when used to express existence |
| | FOREIGN WORD | A non-English word |
| | POSSESSIVE ENDING | The English genitive marker |
| | INFINITY MARKER | The word "to" when used to mark a verb in infinitive |
| | PARTICLE | An uninflected word that typically accompanies another word |
| | QUESTION WORD | A word expressing a question |

Table 1: The nine modern English word classes as given in Blake (1988) and the further PoS-tags adopted from the Penn Treebank tag set (Marcus et al., 1993).

**Extraction of grammatical patterns**   We instantiate our framework by the highlighting function $h_\phi$ that matches certain grammatical patterns, defined as n-grams of PoS tags. In our experimental setup, we adopt the PoS-tagger introduced in Toutanova et al. (2003), which uses the Penn Treebank tag set (Marcus et al., 1993). As the tag set is too extensive, we reduce complexity by consolidating tags into the categories outlined in Table 1. Moreover, we use $n \in \{1, \ldots, 7\}$, resulting in a comprehensive set of 100.004 distinct grammatical features. As similar approaches have previously been successfully applied for authorship attribution (Sidorov et al., 2014), we anticipate these text passages to provide valuable insights into the texts' origin by revealing their grammatical structure.

**Selecting predictive patterns**   Highlighting relevant text passages based on grammar requires understanding which grammatical patterns are informative. This is achieved by training a logistic regression model (Cramer, 2002) with $l_1$-regularization to induce sparsity, making the model reliant only on the most predictive grammatical patterns. Moreover, Miller's law (Miller, 1956) affirms the capacity of most people to retain maximally 9 items in short-term memory. This principle strongly implies that the number of extracted patterns should not significantly surpass this cognitive threshold if interpretability is a desired property. In our experimental setup, we find that relying on 20 grammatical patterns provides a good trade-off between interpretability and performance, which is achieved by adjusting the regularization parameter $\alpha$ from Equation (2).

**Human-in-the-loop**   When assessing whether any text is LLM-generated, text passages matching the extracted grammatical patterns are highlighted and presented to a human who makes the final decision regarding the origin of the text. In order to associate each pattern with either human-written or LLM-generated texts, we refit the logistic regression model on the extracted patterns and assess the sign of the coefficient vector. The resulting, interpretable model can also be evaluated to understand how predictive the information provided to human users truly is. This approach guides decision-makers by directing their attention to the relevant parts of the text but remains interpretable as the final decision is based on specific, verifiable information that is extracted using our model.

## 5 Experiments

In this section, we empirically evaluate the efficacy of our method. First, we show that the extracted grammatical patterns are highly informative and can be applied to detect texts produced by the current state-of-the-art LLMs. Similarly, we demonstrate the robustness of our approach against several evasion strategies. Finally, through a human trial, we demonstrate that our patterns improve the ability of non-experts to recognize LLM-generated text, thus resulting in an interpretable and accurate classification procedure.

**Datasets & metrics** We employ several datasets in our setup: scientific abstracts from arXiv (Kaggle, 2023), social media comments from Reddit (Ethayarajh et al., 2022), CNN news articles (Hermann et al., 2015; See et al., 2017) and Wikipedia entries (Wikimedia Foundation, 2022). In particular, we first obtain human-written samples for each dataset considered. Then, using these as a reference, we query the LLM under consideration to produce similar texts, ensuring alignment in terms of subject matter, literary genres, origin, and length. The specific prompts are given in Appendix B.1. We measure the performance according to the AUROC score, and in Appendix F we additionally report accuracy.

### 5.1 Interpretable detection of LLM text

We first experiment with patterns as features for an interpretable classifier (no human-in-the-loop).

**Detecting different text types** We evaluate Gramtector's ability to identify different types of ChatGPT-generated texts. As seen in Figure 2, Gramtector performs on par with non-interpretable approaches. On all but one dataset, we attain an AUROC score close to 1 when at least 20 features are used; Gramtector even outperforms the RoBERTa and DistilBERT benchmarks on the Wikipedia and arXiv datasets, respectively. Even though we observe a performance decrease on the Reddit dataset, as we show in Appendix E, text length strongly influences performance and for longer Reddit responses Gramtector notably outperforms all non-interpretable benchmarks.

**Detecting different LLMs** We also study Gramtector's performance on texts produced by different state-of-the-art LLMs, in particular, ChatGPT, GPT-4, BARD, and LLAMA-2-70B. For each model, we construct a dataset of arXiv abstracts (Kaggle, 2023). Similar to our results on various textual domains, Gramtector's performance generalizes across LLMs. For ChatGPT, GPT-4, and LLAMA, we obtain AUROC scores close to 1, even outperforming some of the DNN-based methods. The outlier is the dataset containing abstracts produced by BARD where all models perform significantly worse, with Gramtector lagging behind the DNN-based benchmarks. It is possible that BARD-produced texts better resemble their human-written counterparts or that the model uses a more diverse language, making it harder to detect. Nonetheless, we can conclude that Gramtector generalizes to most practical scenarios; to almost all state-of-the-art LLMs and textual domains.

**Robustness.** To evaluate the robustness of Gramtector, we study several common evasion strategies. Specifically, we limit our ablation study to two realistic scenarios where a malicious actor tries to alter the linguistic expression of an LLM either by prompt engineering or paraphrasing sentences containing characteristics associated with the model. More details of the prompts are given in Appendix B.3, which resemble the attacks studied by Sadasivan et al. (2023). As baselines, we employ our framework instantiated with vocabulary or stylometric features, and we limit our investigation to separating human-written and ChatGPT-generated abstracts from arXiv.

In Table 2, we report the accuracy, AUROC score, and true positive ratio (TPR) of all models on the original dataset as well as the datasets containing adversarially constructed LLM samples. Although the model reliant on vocabulary features performs slightly better on the original dataset, Gramtector is significantly more robust in the adversarial setting. Its performance is unchanged under adversarial prompting and only marginally affected by paraphrasing. However, both other detection methods can be trivially evaded using these strategies; paraphrasing is especially effective, reducing both models' TPR from 98% to 4%. It therefore seems like grammatical sentence structure is an intrinsic characteristic of the LLM that is challenging to alter.

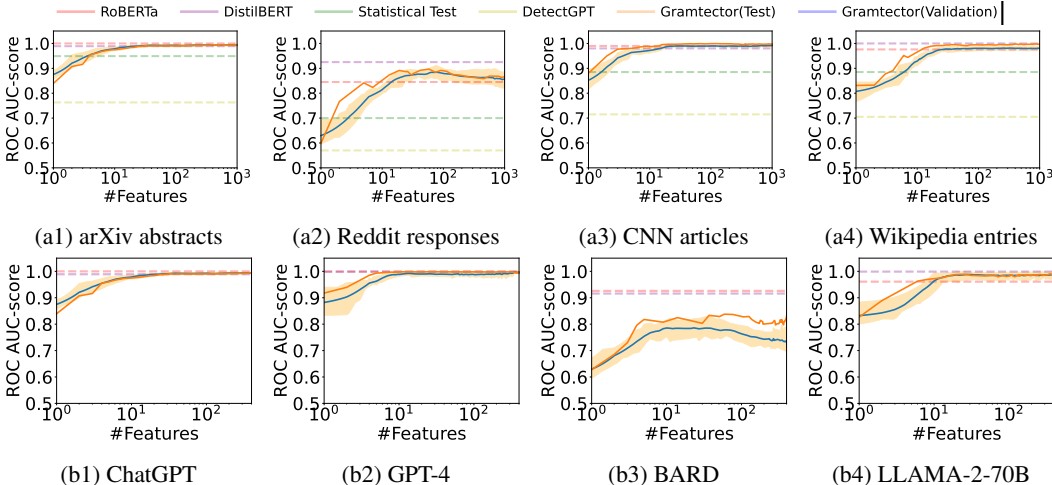

Figure 2: AUROC score dependent on the number of active features. The solid lines indicate the validation and test performance of Gramtector, while the stipulated refer to the benchmarks' utility on the test set. The shaded area marks the 10% and 90% quantiles of Gramtector's AUROC score when employing 10-fold cross-validation.

| | Original Model | | | Adversarial Prompting | | | Adversarial Paraphrasing | | |
|---|---|---|---|---|---|---|---|---|---|
| | Acc. | AUROC | TPR | Acc. | AUROC | TPR | Acc. | AUROC | TPR |
| Vocabulary | **0.970** | **0.997** | **0.980** | 0.900 | 0.983 | 0.840 | 0.500 | 0.546 | 0.040 |
| Stylometric | 0.895 | 0.966 | 0.980 | 0.815 | 0.9111 | 0.820 | 0.425 | 0.588 | 0.040 |
| Gramtector | 0.955 | 0.984 | 0.950 | **0.955** | **0.984** | **0.950** | **0.940** | **0.977** | **0.920** |

Table 2: The efficacy of our framework when instantiated with vocabulary, stylometric, and grammatical (Gramtector) features. The models' accuracy, AUROC score, and true positive ratio (TPR) on the original as well as adversarially constructed datasets are shown.

## 5.2 HUMAN TRIAL

We now describe a human trial where we used our patterns to assist human labelers in the detection.

**Setup.** We assess the efficacy of our approach by replicating plausible scenarios in which non-experts might encounter LLM-generated texts. Specifically, we study the research questions:

**Q1** *Can insights extracted from Gramtector help non-experts recognize LLM-generated texts?*

**Q2** *Which level of AI guidance is most suited to support human decision-making?*

We engage participants in an online study. Each individual is given 10 abstracts which they are asked to classify according to perceived origin. We reveal that 5 are human-authored and 5 are ChatGPT-generated. In the baseline study, this is all the information participants are given. In subsequent experiments, we provide increasingly easier access to the grammatical patterns extracted from Gramtector. Specifically, we employ a tiered approach with three levels:

1. **PoS tagging**: Participants are explained the grammatical patterns and we color each word with its corresponding PoS tag. However, individuals still need to manually search for any matches, which requires comprehending the provided information.

2. **Matched patterns**: All pattern matches are highlighted, but users still have to manually look up in the table whether each pattern is indicative of human or ChatGPT-generated text.

3. **Matched and classified patterns**: Pattern matches associated with human-written texts are highlighted in blue whereas ChatGPT-patterns are colored red. Interpreting this information is then similar to assessing a black-box probability score.

Figure 3 shows the setup for Level 1, PoS tagging, while more details about the setup for all levels are given in Appendix C. Moreover, we ask participants to justify their decisions, allowing us to gauge their level of engagement as expounded on in Appendix D. Specifically, we employ three categories: unengaged responses, engaged responses, and engaged responses employing the provided grammatical patterns. We thus address the trend of using to LLMs complete online surveys (Veselovsky et al., 2023); allowing us to separate hastily completed and thoughtful responses.

We review the developments of QCD multipole expansion and its applications to hadronic transitions and some radiative decays of heavy quarkonia. Theoretical predictions are compsred with updated experimental results.

| ChatPGT | | Human | |
|---|---|---|---|
| DETERMINER NOUN VERB | | DIGIT | |
| this paper presents, this paper investigates, this research contributes | | two, 2, one | |
| PREPOSITION VERB | | VERB ADVERB | |
| by employing, to advancing, by considering | | is not, do not, are not | |
| PRONOUN NOUN | | VERB VERB PREPOSITION | |
| our understanding, our findings, our approach | | is based on, is shown that, is related to | |
| NOUN CONJUNCTION | | PREPOSITION DETERMINER | |
| properties and, accuracy and, behavior and | | of the, in the, to the | |
| DETERMINER NOUN VERB DETERMINER | | MODAL | |
| this paper presents a, this paper investigates the, this paper explores the | | can, may, will | |

(a) Text to classify      (b) Grammatical patterns from Gramtector

Figure 3: The information presented to participants in the human trial at Level 1, PoS tagging. Test takers are given a text (a) to classify according to origin: human-written or ChatGPT-generated. To inform their decision they are given access to the grammatical patterns extracted from Gramtector (b). Furthermore, each word in the text is accentuated with the corresponding PoS tag.

**Human-in-the-loop with Gramtector patterns** We observed in Figure 4a and Table 4 that among engaged participants, most actively employ the provided grammatical patterns, indicating that even non-experts find these insights useful. Notably, also at Level 1, engaged individuals make active use of the information we provide; requiring them to understand the patterns to find matches. Furthermore, as seen in Figure 4b, participants who actively employ grammatical characteristics to detect LLM-produced abstracts, significantly outperform the baseline; increasing the accuracy from 40% to 86% at Level 1. Also at subsequent levels, participants employing the insights extracted from Gramtector better detect LLM-generated texts compared to unengaged participants as well as the baseline, though slightly worse than at Level 1. When studying unengaged participants, their performance steadily increases with easier access to the grammatical characteristics, indicating that they implicitly make use of this information. From Table 3, we observe that the performance of the entire population increases together with the access to the grammatical patterns. Therefore, it indeed seems that the insights encapsulated in Gramtector can be transferred to humans, empowering them to better recognize LLM-generated texts.

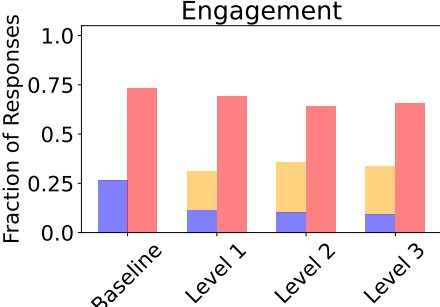

(a) Distribution of engagement among responses for different levels of AI guidance.

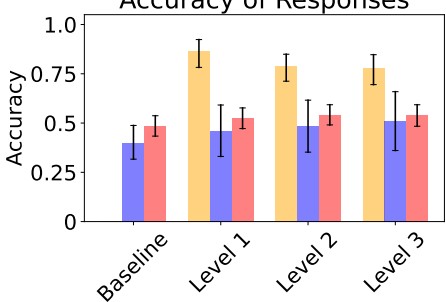

(b) Accuracy of responses conditioned their engagement for different levels of AI guidance.

Figure 4: Trial results by engagement: unengaged (🟥), engaged (🟦), and engaged responses referencing the grammatical patterns (🟨). Error-bars: 5% Clopper & Pearson (1934) confidence interval.

|  | Baseline | Level 1 | Level 2 | Level3 |
|---|---|---|---|---|
| $\hat{p}$ | 0.463 | 0.583 | 0.598 | 0.594 |
| $\hat{p}_{\text{Low}}$ | 0.419 | 0.540 | 0.557 | 0.550 |
| $\hat{p}_{\text{High}}$ | 0.507 | 0.625 | 0.638 | 0.637 |
| True Positive Rate | 0.573 | 0.600 | 0.610 | 0.659 |
| False Positive Rate | 0.537 | 0.417 | 0.402 | 0.406 |
| p-value | N/A | 0.685% | 0.169% | 0.338% |
| Correct Responses | 236 | 309 | 353 | 303 |
| Total Responses | 510 | 530 | 590 | 510 |

Table 3: Estimated accuracy among all participants. $[\hat{p}_{\text{Low}}, \hat{p}_{\text{High}}]$ provides a 5% Clopper & Pearson (1934) confidence interval. The p-value assesses whether $\hat{p}$ is larger at the given level compared to the baseline. LLM-generated texts are considered positive samples for true and false positive rates.

**AI Guidance.** Furthermore, optimal performance among individuals employing the grammatical pattern is attained at Level 1, resulting in a paradoxical situation: increased access to information results in lowered performance. To understand this result, we assess how participants treat the provided information; do they apply the patterns in a black-box fashion, merely counting if they are mostly associated with human-written or LLM-generated texts, or do they contextualize the information? The specific procedure to detect black-box usage is explained in Appendix D. As seen in Table 4, stronger AI guidance correlates with more black-box usage. Therefore, it seems that if the model's predictions are all but directly presented to the user, individuals become overly reliant on AI guidance and do not comprehend and contextualize the provided information. This results in a noticeable performance decrease. Consequently, optimal results seem to be attained in a setting that requires cooperation between humans and AI.

|  | Baseline | Level 1 | Level 2 | Level 3 |
|---|---|---|---|---|
| Black-Box References to Patterns | 0 | 21 | 47 | 87 |
| Engaged Responses Referencing Patterns | 0 | 103 | 150 | 126 |
| Engaged Responses | 135 | 164 | 210 | 173 |
| Total Responses | 510 | 530 | 590 | 510 |
| Pattern Utilization | N/A | 62.8% | 71.4 % | 72.8% |
| Black-Box Pattern Utilization | N/A | 20.4% | 31.3 % | 69.0% |

Table 4: Engagement metrics across the various levels of AI guidance. Pattern utilization is the fraction of engaged responses that reference the provided grammatical patterns. Black-box pattern utilization is the fraction of responses that reference the provided grammatical patterns which do this in a black-box manner.

# 6 CONCLUSION

We introduced Gramtector, a framework for accurately and interpretably detecting LLM-generated texts. Our key insight is to learn grammatical patterns associated with texts produced by an LLM, which can subsequently be employed as identifiable markers of the model. Our experimental evaluation on datasets containing various types of text produced by leading-edge LLMs demonstrated that Gramtector performs on par with state-of-the-art non-interpretable detection methods. Moreover, the method appeared robust against several evasion strategies. A major advantage over prior work is Gramtector's inherent interpretability, allowing its insights to be transferred to humans. Through a human trial, we demonstrated that access to these insights significantly increases human decision-makers' ability to recognize LLM-produced texts, raising their accuracy from 40% to 86%. Our work thereby addresses several key concerns, contributing to the responsible deployment of LLMs.

ETHICS STATEMENT

The proliferation of LLMs raises several societal concerns due to the difficulty of discerning human-authored and LLM-produced texts. Our work aims to mitigate these issues by developing a robust framework for accurately and interpretably detecting LLM-generated texts. Although our framework does not completely remove the issue of false positives, its inherent interpretability allows individuals to understand and refute allegations of LLM use, unlike non-interpretable methods. We also demonstrated that the insights from our method could be transferred to humans, allowing them to better recognize LLM-generated texts. Ethical approval for our study was granted by an independent ethics commission (further details withheld due to double-blind review). Additionally, participants received fair compensation for their contributions, and stringent measures were in place to prevent exposure to harmful content. Overall, our work addresses important societal concerns regarding the widespread use of LLMs in a responsible manner.

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
