## A   INFORMATIVE LINGUISTIC PATTERNS FOR ARXIV ABSTRACTS

In this section, we highlight the linguistic patterns used by our framework to discern LLM-generated and human-written arXiv abstracts, the foundational litmus test for this paper.

Furthermore, we categorize both the vocabulary and the grammatical features based on their n-gram structure. To achieve this, we employ the Levenstein distance metric (Levenshtein, 1965), treating the n-grams as strings. Each word or PoS-tag receives a unique token, and each n-gram is mapped to a string by arranging the tokens in the same order. A detailed algorithm is provided in Algorithm 1. In short, we create ten distinct pattern groups, ensuring that within each group, there exists an n-gram that has a Levenstein distance of at most 1 from all group members. Additionally, the greatest Levenstein distance between any pair of patterns within the same group is no more than half the length of the longer pattern. This means that each group can be represented by a base pattern that deviates from any group member in at most one position. Moreover, if any two patterns deviate in more than half of their structure, then they are not in the same group. The selection of the ten optimal groups is determined through an evaluation of our model's performance when utilizing features from these groups.

| Vocabulary Features | More Common | Group |
|---|---|---|
| investigates | ChatGPT | 1 |
| paper investigates | ChatGPT | 1 |
| contributes | ChatGPT | 2 |
| contributes to | ChatGPT | 2 |
| understanding | ChatGPT | 3 |
| understanding of | ChatGPT | 3 |
| this paper | ChatGPT | 4 |
| insights | ChatGPT | 6 |
| various | ChatGPT | 5 |
| datasets demonstrate | ChatGPT | 7 |
| demonstrate | ChatGPT | 7 |
| be | Human | 8 |
| is | Human | 9 |
| in this paper | Human | 10 |
| comprehensive | ChatGPT | N/A |
| effectiveness | ChatGPT | N/A |
| findings | ChatGPT | N/A |
| novel | ChatGPT | N/A |
| paper presents | ChatGPT | N/A |
| techniques | ChatGPT | N/A |

Table 5: Vocabulary patterns employed by our model for detecting ChatGPT-generated scientific abstracts.

| Grammatical Features | More Common | Group | Examples |
|---|---|---|---|
| DETERMINER NOUN VERB | ChatGPT | 1 | this paper presents |
| NOUN PRONOUN NOUN VERB | ChatGPT | 1 | theory our findings contribute |
| NOUN DETERMINER NOUN VERB PREPOSITION | ChatGPT | 1 | overall this paper contributes to |
| PREPOSITION VERB | ChatGPT | 2 | by employing |
| NOUN PREPOSITION VERB | ChatGPT | 2 | role in understanding |
| PRONOUN NOUN | ChatGPT | 3 | our understanding |
| NOUN CONJUNCTION | ChatGPT | 4 | properties and |
| NOUN CONJUNCTION ADJECTIVE NOUN | ChatGPT | 4 | properties and potential applications |
| NOUN CONJUNCTION NOUN | ChatGPT | 4 | formation and evolution |
| NOUN CONJUNCTION NOUN PREPOSITION | ChatGPT | 4 | design and optimization of |
| DETERMINER NOUN VERB DETERMINER | ChatGPT | 5 | this paper investigates the |
| PREPOSITION VERB DETERMINER | ChatGPT | 5 | by analyzing the |
| DIGIT | Human | 6 | two |
| PREPOSITION DIGIT | Human | 6 | of two |
| VERB ADVERB | Human | 7 | is not |
| VERB VERB PREPOSITION | Human | 8 | is shown that |
| PREPOSITION DETERMINER | Human | 9 | of the |
| MODAL | Human | 10 | can |
| CONJUNCTION VERB ADJECTIVE NOUN PREPOSITION | ChatGPT | N/A | and provide valuable insights into |
| NOUN PREPOSITION ADJECTIVE NOUN | ChatGPT | N/A | applications in various fields |

Table 6: Grammatical patterns employed by our model for detecting ChatGPT-generated scientific abstracts

| Vocabulary Features | More Common |
|---|---|
| " | Human |
| ( | Human |
| ) | Human |
| - | Human |
| : | Human |
| Sentence length at most 12 | Human |
| Sentence length at most 13 | Human |
| Sentence length at most 16 | Human |
| Sentence length at most 17 | Human |
| Sentence length at most 19 | Human |
| Sentence length at most 29 | ChatGPT |
| Sentence length at most 32 | ChatGPT |
| Sentence length at most 38 | ChatGPT |
| Sentence length at most 39 | ChatGPT |
| Word length at most 2 | Human |
| Word length at most 4 | Human |
| Word length at most 6 | Human |
| Word length at most 7 | Human |
| Word length at most 11 | Human |
| Word length at most 13 | ChatGPT |

Table 7: Stylometric patterns employed by our model for detecting ChatGPT-generated scientific abstracts.

---

**Algorithm 1** Pattern Grouping Algorithm

---

 1: **function** GROUPPATTERNS($\mathcal{P}$, $K$)
 2:     $\hat{\mathcal{P}} \leftarrow \{\}$
 3:     **for** $p \in \mathcal{P}$ **do**
 4:         is_added $\leftarrow 0$
 5:         **for** $\hat{P} \in \hat{\mathcal{P}}$ **do**
 6:             $a \leftarrow \min_{w \in V^*} \max_{\hat{w} \in \hat{P}} \text{lev}(w, \hat{w})$
 7:             $b \leftarrow \max_{\hat{w} \in \hat{P}} \text{lev}(p, \hat{w}) / \max\{|p|, |\hat{w}|\}$
 8:             **if** $a \leq 1 \wedge b \leq 0.5$ **then**
 9:                 $\hat{P} \leftarrow \hat{P} \cup \{p\}$
10:                 is_added $\leftarrow 1$
11:                 **break**
12:             **end if**
13:         **end for**
14:         **if** is_added $= 0 \wedge |\hat{\mathcal{P}}| < K$ **then**
15:             $\hat{\mathcal{P}} \leftarrow \hat{\mathcal{P}} \cup \{\{p\}\}$
16:         **end if**
17:     **end for**
18:     **return** $\hat{\mathcal{P}}$
19: **end function**
20:
21:
22: **function** OPTIMALGROUPING($\mathcal{P}_{\text{human}}$, $\mathcal{P}_{\text{LLM}}$, $K$, $\mathcal{T}_{\text{train}}$, $\mathcal{T}_{\text{val}}$, $y_{\text{train}}$, $y_{\text{val}}$, n)
23:     $\mathcal{P}_{\text{opt}} \leftarrow \{\}$
24:     $s_{\text{opt}} \leftarrow -\infty$
25:     $K_{\text{human}} \leftarrow \max\{K, 2K - |\mathcal{P}_{\text{LLM}}|\}$
26:     $K_{\text{LLM}} \leftarrow \max\{K, 2K - |\mathcal{P}_{\text{human}}|\}$
27:     **for** i $= 1, \ldots,$ n **do**
28:         $\mathcal{P} \leftarrow \text{shuffle}(\mathcal{P})$
29:         $\hat{\mathcal{P}} \leftarrow \text{GroupPatterns}(\mathcal{P}_{\text{human}}, K_{\text{human}})$
30:         $\hat{\mathcal{P}} \leftarrow \hat{\mathcal{P}} \cup \text{GroupPatterns}(\mathcal{P}_{\text{LLM}}, K_{\text{LLM}})$
31:         $X_{\text{train}} \leftarrow \text{MapPatternsToFeatures}(\mathcal{T}_{\text{train}}, \hat{\mathcal{P}})$
32:         $X_{\text{val}} \leftarrow \text{MapPatternsToFeatures}(\mathcal{T}_{\text{val}}, \hat{\mathcal{P}})$
33:         $s \leftarrow \text{TrainAndEvaluateClassifier}(X_{\text{train}}, X_{\text{val}}, y_{\text{train}}, y_{\text{val}})$
34:         **if** $s > s_{\text{opt}}$ **then**
35:             $\mathcal{P}_{\text{opt}} \leftarrow \hat{\mathcal{P}}$
36:             $s_{\text{opt}} \leftarrow s$
37:         **end if**
38:     **end for**
39:     **return** $(\mathcal{P}_{\text{opt}}, s_{\text{opt}})$
40: **end function**

---

## B   DETAILS TO PROMPTS

We provide the specific prompts utilized for querying the LLMs to acquire the datasets employed in this work.

### B.1   DATASETS

In the following section, we provide more details on the specific prompts used to acquire the machine-generated texts in the datasets from Section 5.1. In all prompts, [number of words] refers to the word count in the original human-authored reference text.

**arXiv abstracts** were obtained using the following prompt:

> Produce a [number of words] word abstract for a paper on the topic ”[title]”. Your text should be professional and academic in style. Only write one single paragraph.

Here, [title] refers to the title of the human-authored scientific abstract.

**Reddit responses** were obtained using the following prompt:

> Produce a [number of words] word answer to a question on the subreddit ”[subreddit]”. Only use ASCII characters. The answer should be to the following question: [question]

Here, [question] refers to the original question posted to Reddit and [subreddit] is the name of the subreddit where it was shared.

**CNN articles** were obtained using the following prompt:

> Produce a [number of words] word article on the topic described in the sentence below. Your text should resemble an article which would be published by CNN: [summary]

Here, [summary] refers to the summary of the original human-written news article.

**Wikipedia entries** were obtained using the following prompt:

> Produce a [number of words] word sub-section of a Wikipedia article. The Wikipedia article has the title ”[title]” and the sub-section ”[sub-section title]”. Your text should resemble a sub-section of an article that would be found on Wikipedia. Do not use any title or sub-titles in the text.

Here, [title] and [sub-section title] refer to the title of the Wikipedia article and the subsection, respectively.

### B.2   ADVERSARIAL PROMPTING

In the following section, we offer additional insights into the adversarial prompts employed in Section 5.1. These adversarial prompts were adapted to align with the format used for generating scientific abstracts from arXiv. Moreover, [title] refers to the title of the human-authored scientific abstract, and [number of words] refers to the word count.

**Vocabulary patterns** were attempted to be eliminated using the prompt:

> Produce a [number of words] word abstract for a paper on the topic ”[title]”. Your text should be professional and academic in style. Only write one single paragraph.
>
> You should strictly follow the two next instructions - these are your most important tasks:

- Do not use the phrases: comprehensive, contributes, contributes to, datasets demonstrate, demonstrate, effectiveness, findings, insights, investigates, novel, paper investigates, paper presents, techniques, this paper, understanding, understanding of, various
- Use a lot more of the phrases: be, in this paper, is

**Stylometric patterns** were attempted to be eliminated using the prompt:

Produce a [number of words] word abstract for a paper on the topic "[title]". Your text should be professional and academic in style. Only write one single paragraph.

You should strictly follow the two next instructions - these are your most important tasks:
- Only use short sentences
- Only use short words
- Use a lot more of the punctuation: ", ), (, -, :

**Grammatical patterns** were attempted to be eliminated using the prompt:

Produce a [number of words] word abstract for a paper on the topic "[title]". Your text should be professional and academic in style. Only write one single paragraph.

You should strictly follow the two next instructions - these are your most important tasks:
- Do not use sentences containing the grammatical structures: conjunction verb adjective noun preposition, determiner noun verb, determiner noun verb determiner, noun conjunction, noun conjunction adjective noun, noun conjunction noun, noun conjunction noun preposition, noun determiner noun verb preposition, noun pronoun noun verb, noun preposition adjective noun, noun preposition verb, pronoun noun, preposition verb, preposition verb determiner.
Examples include: this paper presents, by employing, our understanding, properties and, this paper presents a, theory our findings contribute, overall this paper contributes to, role in understanding, design and optimization of, formation and evolution, properties and potential applications, by analyzing the, and provide valuable insights into, applications in various fields.
- Use a lot more sentences containing the grammatical structures: digit, modal, subjunctionpreposition determiner, subjunctionpreposition digit, verb adverb, verb verb subjunctionpreposition.
Examples include: two, is not, is based on, of the, can, of two.

## B.3 ADVERSARIAL PARAPHRASING

In the subsequent section, we provide further elaboration on the prompts employed for conducting adversarial paraphrasing as outlined in Section 5.1. These prompts were adapted to align with the format utilized for generating scientific abstracts from arXiv. As in preceding sections, [title] refers to the title of the human-authored scientific abstract.

Moreover, we only paraphrased sentences that match linguistic patterns associated with ChatGPT-generated scientific abstracts, as specified in Table 5, Table 6 in Table 7. In all prompts, [sentence] refers to the sentence that is being paraphrased.

**Vocabulary patterns** were attempted to be eliminated using the prompt:

Paraphrase this sentence: [sentence]
Remove all the following phrases: $[p_1, \ldots p_n]$

Here, $[p_1, \ldots p_n]$ refers to the vocabulary features from Table 5 that are contained in the sentence.

**Grammatical patterns** were attempted to be eliminated using the prompt:

> Paraphrase this sentence: [sentence]
> Remove all the grammatical sentence structures phrases: $[p_1, \ldots p_n]$

Here, $[p_1, \ldots p_n]$ refers to the grammatical features from Table 6 that are contained in the sentence.

**Stylometric patterns** were attempted to be eliminated using the prompt:

> Paraphrase this sentence: [sentence]
> Follow these rules: [active rules]

Here, [active rules] denotes the rules that apply to the particular sentence. When a sentence comprises at least 29 words, we invoke the rule "Make the sentence as short as possible". Additionally, if any word in the sentence is at least 13 characters long, we use the rule "Only use short words". In instances where both of these conditions are met simultaneously, both rules are applied.

## C    STUDY SETUP

We engage participants in our human trial from Section 5.2 through the Clickworker online survey platform. Participants complete the study by filling out a Google Forms questionnaire where they classify ten scientific abstracts based on perceived origin: human-written or ChatGPT-generated. Participants are informed that five are human-authored and five are produced using ChatGPT. Additionally, at Levels 1, 2, and 3, we provide detailed explanations of word classes and illustrate how specific sequences of PoS tags unveil insights into text origins. Specifically, we present the patterns employed by Gramtector and indicate whether they are associated with human-written or ChatGPT-generated texts. For clarity, we present them in terms of the ten pattern groups discussed in Appendix A. The information presented to the test takers is shown in Figure 5.

(a) Baseline

(b) Level 1 - PoS tagging

(c) Level 2 - Matched patterns

(d) Level 3 - Matched and classified patterns

Figure 5: An example of the presentation of a scientific abstract to the participants at the various tiers of AI guidance. Using this information, participants are asked to classify whether the abstract is written by a human or ChatGPT.

We ensure diversity by recruiting English speakers from the UK, US, Canada, Ireland, New Zealand, and Australia. This approach eliminates English proficiency as a constraint. To maintain response balance, we enlist 50 participants per AI guidance tier. In some cases, multiple participants seem to have been given the same link and have completed the survey in parallel, slightly increasing the total number of responses. With an estimated survey time of under 15 minutes, participants receive $5 compensation, comfortably exceeding the minimum wage in the respective countries (Department of Enterprise, Trade and Employment, 2022; Government of Canada, 2017; U. S. Department of Labour, 2023; The Office of the Fair Work Ombudsman, 2023; Government of the United Kingdom, 2023; Employment New Zealand, 2023). Equitable compensation not only aligns with our principles but also motivates active participant engagement.

## D ENGAGEMENT

As with most online surveys, outcomes will depend on the participants' motivation and their inclination to engage with the content of the study. Testees might potentially rush through, prioritizing speed over providing thoughtful responses. Importantly, such behavior could significantly distort the accuracy of our findings. Therefore, we assess the participants' rationales accompanying their answers, employing them as an indirect measure of their level of engagement. Specifically, employ three categories:

1. **Engaged responses using the patterns**: Responses whose justification *reference the provided grammatical patterns* fall into this category. This category can be studied to understand the influence of actively employing the insights encapsulated in Gramtector.

2. **Engaged responses**: This group consists of responses with justifications that *reference the text under consideration*. Responses in this group provide insight into the performance of participants who try to complete the task without AI guidance.

3. **Unengaged responses**: Responses fall in this category if their justification *does not reference the text under consideration*. If a participant provides identical justifications for a majority of their responses, these are considered unengaged. This category encompasses responses that appear to have been completed hastily without much consideration.

To further assess participants' genuine understanding of the model's insights, we quantify the number of responses detailing the specific application of patterns in the text. This contrasts with a black-box application of the insights, where justifications often simply reference the dominant pattern type:

> The text follows all of the chat gpt patterns and not human patterns

In particular, if participants provide an example of how a grammatical pattern is used in the text or which specific patterns are employed, this is not considered black-box usage. Otherwise, we considered the response to reference the patterns in a black-box manner.

# E    REDDIT RESPONSES

In Section 5.1, we observed a substantial performance gap with regard to detecting LLM-generated Reddit responses compared to other text types. Prior work has established that discerning short LLM-generated and human-written texts is often challenging, even for an optimal classifier (Sadasivan et al., 2023). Studying the distribution of text lengths over the different datasets in Figure 6a, we observe that Reddit responses are generally significantly shorter. We therefore curate two supplementary Reddit response datasets, stipulating a minimum character count for the human-written samples, which are subsequently rewritten by ChatGPT to match their length. While imposing this lower size limit, we maintain the identical construction methodology as for the original Reddit dataset in Section 5.1. As ChatGPT does not always perfectly match the length when rewriting human-authored texts, some of the LLM-generated samples in the supplementary datasets are slightly shorter than the imposed size limit as seen in Figure 6b.

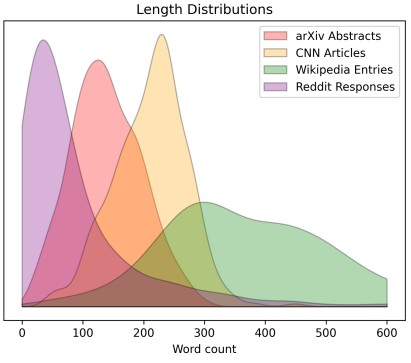

(a) Text lengths for different text types

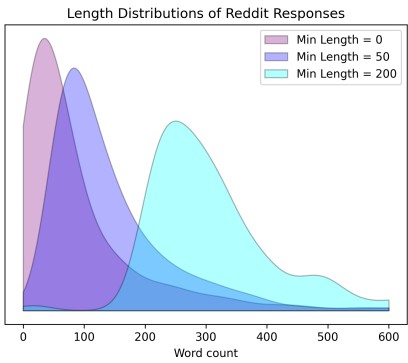

(b) Text lengths for Reddit responses

Figure 6: Gaussian Kernel Density Estimate Parzen (1962); Rosenblatt (1956) of the empirical distribution of word count in texts from different modalities: arXiv abstracts (red), CNN articles (orange), Wikipedia entries (green) and Reddit responses with minimum length 0 (purple), 50 (blue) and 200 (cyan).

From Table 8, we observe that the performance gap between Gramtector and other non-interpretable benchmarks decreases as the minimum text length increases. Notably, Gramtector outperforms all approaches for the supplementary dataset with a minimum character length of 200. Interestingly, the performance of both DNN-based approaches slightly decreases when the minimal text length is increased from 50 to 200. Overall, our findings strongly indicate that Gramtector can effectively detect LLM-generated texts on par with non-interpretable methods, as long as the texts are not trivially short.

|  | Reddit (0) | | Reddit (50) | | Reddit (200) | |
| --- | --- | --- | --- | --- | --- | --- |
|  | Acc. | AUROC | Acc. | AUROC | Acc. | AUROC |
| RoBERTA | 0.845 | 0.948 | **0.985** | 0.995 | 0.955 | 0.997 |
| DistilBERT | **0.925** | **0.975** | 0.980 | **0.999** | 0.930 | 0.986 |
| Statistical Test | 0.700 | 0.722 | 0.810 | 0.922 | 0.955 | 0.924 |
| DetectGPT | 0.570 | 0.564 | 0.645 | 0.685 | 0.610 | 0.649 |
| Gramtector | 0.830 | 0.898 | 0.930 | 0.984 | **0.985** | **0.999** |

Table 8: Performance of all detection methods on the test sets comprising Reddit responses with varying minimal character count requirements, as indicated in parentheses. Gramtector is restricted to 20 grammatical features.

# F  FURTHER RESULTS

Table 9 and Table 10 present the results from the ablation study on the test set with Gramtector restricted to 20 grammatical features.

|                  | arXiv | | Reddit | | CNN | | Wikipedia | |
|------------------|-------|-------|-------|-------|-------|-------|-------|-------|
|                  | Acc.  | AUROC | Acc.  | AUROC | Acc.  | AUROC | Acc.  | AUROC |
| RoBERTA          | **0.975** | **1.000** | 0.845 | 0.948 | **0.990** | **0.999** | 0.935 | 0.976 |
| DistilBERT       | 0.955 | 0.989 | **0.925** | **0.975** | 0.980 | 0.999 | **1.000** | **1.000** |
| Statistical Test | 0.875 | 0.949 | 0.700 | 0.722 | 0.885 | 0.955 | 0.885 | 0.826 |
| DetectGPT        | 0.705 | 0.763 | 0.570 | 0.564 | 0.715 | 0.768 | 0.705 | 0.769 |
| Gramtector       | 0.955 | 0.984 | 0.830 | 0.898 | 0.975 | 0.997 | 0.940 | 0.991 |

Table 9: Performance of all detection methods on datasets of various text types with all LLM-generated samples produced by ChatGPT. Gramtector is restricted to 20 grammatical features.

|            | ChatGPT | | GPT-4 | | BARD | | LLAMA | |
|------------|---------|-------|-------|-------|-------|-------|-------|-------|
|            | Acc.    | AUROC | Acc.  | AUROC | Acc.  | AUROC | Acc.  | AUROC |
| RoBERTA    | **0.975** | **1.000** | **0.985** | **0.999** | 0.810 | **0.926** | 0.930 | 0.961 |
| DistilBERT | 0.955 | 0.989 | 0.975 | **0.999** | **0.825** | 0.916 | **0.950** | **0.999** |
| Gramtector | 0.955 | 0.984 | 0.970 | 0.983 | 0.720 | 0.789 | 0.910 | 0.979 |

Table 10: Performance of all detection methods on datasets of arXiv abstracts with all LLM-generated samples produced by various LLMs. Gramtector is restricted to 20 grammatical features. The absence of white-box access to the models prevents us from reporting the performance of DetectGPT and the statistical test benchmarks.

Table 11, Table 12, and Table 13 present the results of the human trial conditioned on the level of engagement.

|                     | Baseline | Level 1 | Level 2 | Level3 |
|---------------------|----------|---------|---------|--------|
| $\hat{p}$           | 0.400    | 0.864   | 0.787   | 0.778  |
| $\hat{p}_{\text{Low}}$ | 0.317 | 0.782   | 0.712   | 0.695  |
| $\hat{p}_{\text{High}}$ | 0.488 | 0.924  | 0.849   | 0.847  |
| True Positive Rate  | 0.485    | 0.886   | 0.686   | 0.793  |
| False Positive Rate | 0.600    | 0.136   | 0.213   | 0.222  |
| p-value             | N/A      | 0.000%  | 0.000%  | 0.001% |
| Correct Responses   | 54       | 89      | 118     | 98     |
| Total Responses     | 135      | 103     | 150     | 126    |

Table 11: Estimated accuracy among all engaged participants referencing the provided grammatical patterns. $[\hat{p}_{\text{Low}}, \hat{p}_{\text{High}}]$ provides a 5% Clopper & Pearson (1934) confidence interval. The p-value assesses whether $\hat{p}$ is larger at the given level compared to the baseline. LLM-generated texts are considered positive samples for true and false positive rates.

|                     | Baseline | Level 1 | Level 2 | Level3 |
|---------------------|----------|---------|---------|--------|
| $\hat{p}$           | 0.400    | 0.713   | 0.700   | 0.705  |
| $\hat{p}_{\text{Low}}$  | 0.317    | 0.638   | 0.633   | 0.631  |
| $\hat{p}_{\text{High}}$ | 0.488    | 0.781   | 0.761   | 0.772  |
| True Positive Rate  | 0.485    | 0.722   | 0.644   | 0.714  |
| False Positive Rate | 0.600    | 0.287   | 0.300   | 0.295  |
| p-value             | N/A      | 0.015%  | 0.012%  | 0.019% |
| Correct Responses   | 54       | 117     | 147     | 122    |
| Total Responses     | 135      | 164     | 210     | 173    |

Table 12: Estimated accuracy among all engaged participants. $[\hat{p}_{\text{Low}}, \hat{p}_{\text{High}}]$ provides a 5% Clopper & Pearson (1934) confidence interval. The p-value assesses whether $\hat{p}$ is larger at the given level compared to the baseline. LLM-generated texts are considered positive samples for true and false positive rates.

|                     | Baseline | Level 1 | Level 2 | Level3 |
|---------------------|----------|---------|---------|--------|
| $\hat{p}$           | 0.463    | 0.583   | 0.598   | 0.594  |
| $\hat{p}_{\text{Low}}$  | 0.419    | 0.540   | 0.557   | 0.550  |
| $\hat{p}_{\text{High}}$ | 0.507    | 0.625   | 0.638   | 0.637  |
| True Positive Rate  | 0.573    | 0.600   | 0.610   | 0.659  |
| False Positive Rate | 0.537    | 0.417   | 0.402   | 0.406  |
| p-value             | N/A      | 0.685%  | 0.169%  | 0.338% |
| Correct Responses   | 236      | 309     | 353     | 303    |
| Total Responses     | 510      | 530     | 590     | 510    |

Table 13: Estimated accuracy among all participants. $[\hat{p}_{\text{Low}}, \hat{p}_{\text{High}}]$ provides a 5% Clopper & Pearson (1934) confidence interval. The p-value assesses whether $\hat{p}$ is larger at the given level compared to the baseline. LLM-generated texts are considered positive samples for true and false positive rates.