# OpenReview forum: "Human-in-the-loop Detection of AI-generated Text via Grammatical Patterns"
_ICLR.cc/2024/Conference — Submitted to ICLR 2024_

### Official Review · Reviewer_1GBn · 2023-10-30

**Soundness:** 3 good
**Presentation:** 2 fair
**Contribution:** 2 fair
**Rating:** 3
**Confidence:** 3

**Summary:**

This work leverages grammatical patterns in text for human-int-the-loop AI text detection. The authors claim that this makes their work novel since it adds interpretability and improves the accuracy of detection. They compare their method and show on-par performance against some of the existing text detectors in some settings. They also provide some robustness analysis of their detection technique.

**Strengths:**

- The detector provides interpretability for detection.
- On-par performance with BERT-based detectors in some settings.
- Non-expert humans find the insights extracted from Gramtector helpful.

**Weaknesses:**

- No comparisons with watermarking. (see questions for details)
- The performance of Gramtector is worse than the DistilBert detector for BARD generations or Reddit responses.
- The presentation of the paper can be improved. For e.g., the name Gramtector is introduced without any context and without making clear it's the proposed method. The explanation for what a "group" is in Table 5 is not clear. Algorithm 1 could have been made clearer with some descriptions.
- The robustness analysis in Sec 5.1 is not convincing. (see questions for details)

**Questions:**

1. I find that the important strength of the work is its interpretability. It's important to clarify why one should use Gramtector and not watermarking in Kirchenbauer et al. (2023), given that watermarking has high accuracy? Watermarking is interpretable as well. The number of green tokens and the z-test gives an interpretable score for detection. It can also highlight snippets in text that might be potentially AI-generated. I do not agree with the authors mentioning that "all of these methods do not provide an explanation of why some text has been classified as AI" in the introduction.
2. For adversarial paraphrasing in B.2, how are the lists of "do not use the phrases" and "use a lot more of the phrases" generated? Are these the top features? How many features does the detector use in this setting?
3. The authors mention, "we query the LLM under consideration to produce similar texts". In B.1, it's shown that for arXiv abstracts, the LLMs are only prompted to generate with high-level instructions and a paper title. How would the detection performance change if the LLMs were prompted to write in a particular style? For example, what if the LLMs were given an example human abstract in the prompt and was instructed to generate another one with the same style?
4. In table 2, why are the values of Gramtector for "Original model" and "Adversarial prompting" the same? There is no discussion on why the AUROC is maintained after adversarial paraphrasing.

---

### Official Review · Reviewer_sRmw · 2023-10-31

**Soundness:** 3 good
**Presentation:** 2 fair
**Contribution:** 2 fair
**Rating:** 5
**Confidence:** 3

**Summary:**

The paper proposes a new technique, called gramtect to segregate AI-generated from Human-generated text. Gramtect is an $l_1$ regularized model that works on extracted part-of-speech tagged text. The paper then uses the model in conjunction with human labelers to deliver state-of-the-art performance on classifying human-generated abstracts from ChatGPT-generated abstracts. The paper also investigates the accuracy of gramtect under certain adversarial conditions, like having the LLM not use certain words or phrases, to show the robustness of the model.

**Strengths:**

The paper has strengths in its empirical validation and insight about the text produced by LLMs. First, the empirical validation is quite thorough. The paper investigates the use of grammatical features for LLM-generated text classification not only across a number of realistic settings for artificial text generation (e.g., Wikipedia edits, social media posts, news articles, etc.) but also does empirical validation in realistic adversarial settings. The paper also conducts an actual human-in-the-loop survey, with a number of types of interactions with the data, and shows the best-performing way of combining the machine-generated features with human labelers (i.e., giving grammatical patterns, but not the full classification results)

The insight in the paper about the relative immutability of LLM grammatical patterns, versus the use of certain words or phrases is also important. It is this observation that is central to the proposed method paper and does work quite well and quite robustly in practice.

**Weaknesses:**

The paper does have a few weaknesses. First, the technical contribution of the paper is quite small. The paper essentially only proposes a standard supervised machine learning model that works on existing part-of-speech tagging algorithms. While this type of approach is powerful when combined with human labelers, it is not, by itself a significant contribution. Furthermore, from the human trials, the fact that there is better performance by only exposing human labelers to the part-of-speech tags versus to the actual labels from the gramtect model further decreases the utility of the proposed technical contribution. Perhaps greater performance and a more technical contribution could have been achieved by doing something like having gramtect label the sequence and then an LLM provides an explanation as to why the sequence was labeled as it was to the user.

Also, there are some clarity issues with the paper. Most notably, I cannot find details of which features the regularized model selected nor how the model was trained (e.g., cross-validation, which datasets, etc.)

**Questions:**

I have a number of questions from reading the manuscript, which I order from most difficult to answer to least.

-	How would this method deal with more complex types of text that is a combination of human and AI generation? For example, people will commonly write a rough draft of a text, especially something like an e-mail, and then have the LLM clean up any grammar or clarity issues. Would something like this be counted as LLM-generated text, or not? And, what would this method do in this case? Another example would be having an LLM write the first draft and then a human goes back through the text to clean up parts they don’t particularly like or add to what the LLM produced. Such a practice is used in more creative writing.
-	How is the text handled for classification with gramtect? In section 5.1, it looks like all of the text of a sample is fed to the model and given one classification. In section 5.2, its not clear if individual sentences are fed to the model for a classification or if there is some kind of text matching between features learned in 5.1 and the test text given to human labelers.
-	What is $TV()$ in the lit review?
-	Is the 100.004 distinct grammatical features on page 5 meant to be 100,004 distinct features?

---

### Official Review · Reviewer_G9ty · 2023-10-31

**Soundness:** 2 fair
**Presentation:** 2 fair
**Contribution:** 2 fair
**Rating:** 3
**Confidence:** 3

**Summary:**

This paper presents a human-in-the-loop approach to interpretably identify AI-generated text. This approach uses POS tagging and feature ranking to help users detect AI-generated text. Evaluations show that this framework improves human's detection accuracy. In addition, robustness analysis shows that this framework is robust to different prompts and LLMs.

**Strengths:**

S1: The paper addresses a very timely research problem on detecting AI-generated text.

S2: I like the robustness evaluation and user study. The study provides some promising results of the mixed-initiative approach in AI-generated text detection.

S3: Overall, the paper is well-written and easy to follow.

**Weaknesses:**

## Major weaknesses

W1: The motivation for interpretability in AI-generated text detection can be improved. The introduction (and the paper) would be stronger if it could clarify why we need interpretability in AI-generated text detection. Currently, the introduction only states that "[…] non-interpretability mean that a large number of innocent people will be accused of submitting AI-written text while receiving no explanation for this decision."

W2: The proposed method that uses PoS tagging to help humans detect AI-generated text is not particularly novel.

W3: The evaluation can be improved.
1. The main contribution of this work is a human-in-the-loop approach to detect AI-generated text, but the experiments in Section 5.1 do not involve this approach.
2. The research questions in Section 5.2 are interesting, but they do not really evaluate the effectiveness of the proposed approach.
3. The user study misses important details, such as number of participants and requirement method.

## Minor weaknesses

M1: I recommend referring to the dataset of human-written samples as "reference dataset" instead of "training dataset" to avoid confusion with the original dataset of the LLMs.

M2: The human-in-the-loop approach for detecting LLM-generated text is not novel. For example, [1] and [2] have proposed similar mix-initiative methods. ([1] and [2] are cited in the paper)

M3: Is $e_i$ natural language explanations in equation 3?

M4: "Gramtector" is not defined (section 4).

## References

[1] Gehrmann, Sebastian, Hendrik Strobelt, and Alexander M. Rush. "Gltr: Statistical detection and visualization of generated text." arXiv preprint arXiv:1906.04043 (2019).

[2] Weng, Luoxuan, Minfeng Zhu, Kam Kwai Wong, Shi Liu, Jiashun Sun, Hang Zhu, Dongming Han, and Wei Chen. "Towards an Understanding and Explanation for Mixed-Initiative Artificial Scientific Text Detection." arXiv preprint arXiv:2304.05011 (2023).

**Questions:**

Q1: What is "Gramtector"?

Q2: How many participants were in the user study in Section 5.2? How were they recruited?

---

### Official Review · Reviewer_7kmh · 2023-11-02

**Soundness:** 3 good
**Presentation:** 3 good
**Contribution:** 3 good
**Rating:** 3
**Confidence:** 3

**Summary:**

This paper proposes a method for AI generation detection. It utilizes the linguistic pattern ( part of speech tagging ) as input features to train a linear classifier, and further using the linear classifier to help human annotators to tell apart human text and machine generated ones. Experiments on multiple LLM and multiple domain shows that the performance of linear classifier is on par with DNN methods.

**Strengths:**

+ The paper introduce a simple but effective method for AI generation detection.
+ The writing is clear and easy to follow.
+ Compared with previous method, grammatical pattern proposed by the work is more interpretable.

**Weaknesses:**

+ I wish I could only blame myself but I don't really understand why we need a human-in-loop method, especially when the performance human-in-loop method is on par with a simple DNN method.
+ Some important details are missing. For example, how are the machine-generated Arxiv datasets created in detail? Do you use greedy decoding or top-k/top-p sampling? Moreover, in the human trial, what is the criteria for choosing human participant? What is the point of revealing the distribution of the source ($5$ are human-authored and $5$ are ChatGPT-generated).
+ The baseline methods compared are relatively weak. It is recommended that some recent methods like [1].



[1]Paraphrasing evades detectors of AI-generated text, but retrieval is an effective defense

**Questions:**

See the weakness above.

---

### Meta-Review · Area_Chair_VnN3 · 2023-12-07

**Metareview:**

All reviewers agree that the paper is below the bar. Authors haven't responded to the comments during the rebuttal period.

**Justification For Why Not Higher Score:**

See the above.

**Justification For Why Not Lower Score:**

NA

---

### Decision · Program_Chairs · 2024-01-16

Reject